# Expanding Phenotype of Poirier–Bienvenu Syndrome: New Evidence from an Italian Multicentrical Cohort of Patients

**DOI:** 10.3390/genes13020276

**Published:** 2022-01-30

**Authors:** Alessandro Orsini, Andrea Santangelo, Francesca Bravin, Alice Bonuccelli, Diego Peroni, Roberta Battini, Thomas Foiadelli, Veronica Bertini, Angelo Valetto, Michele Iacomino, Vincenzo Nigro, Anna Laura Torella, Marcello Scala, Valeria Capra, Maria Stella Vari, Anna Fetta, Veronica Di Pisa, Francesca Montanari, Roberta Epifanio, Paolo Bonanni, Roberto Giorda, Francesca Operto, Grazia Pastorino, Esra Sarigecili, Esra Sardaroglu, Cetin Okuyaz, Sevgan Bozdogan, Luciana Musante, Flavio Faletra, Caterina Zanus, Alessandro Ferretti, Federico Vigevano, Pasquale Striano, Duccio Maria Cordelli

**Affiliations:** 1Pediatric Neurology, Pediatric Department, Santa Chiara University Hospital, Azienda Ospedaliero Universitaria Pisana, 56126 Pisa, Italy; aorsini.md@gmail.com (A.O.); francescabravin@gmail.com (F.B.); dott.bonuccelli@alice.it (A.B.); diego.peroni@unipi.it (D.P.); 2Department of Clinical and Experimental Medicine, University of Pisa, 56126 Pisa, Italy; rbattini@fsm.unipi.it; 3Stella Maris Foundation, IRCCS, 56126 Calambrone, Italy; 4Pediatric Clinic, Department of Clinical-Surgical, Diagnostic and Pediatric Sciences, IRCCS Policlinico San Matteo Foundation—University of Pavia, 27100 Pavia, Italy; thomas.foiadelli@gmail.com; 5Cytogenetics Unit, Santa Chiara University Hospital, Azienda Ospedaliero Universitaria Pisana, 56126 Pisa, Italy; v.bertini@ao-pisa.toscana.it (V.B.); a.valetto@ao-pisa.toscana.it (A.V.); 6Unit of Medical Genetics, IRCCS Istituto Giannina Gaslini, 16147 Genova, Italy; m.iacomino87@gmail.com (M.I.); valeriacapra@gaslini.org (V.C.); 7Medical Genetics, Department of Biochemistry, Biophysics and General Pathology University of Campania, Luigi Vanvitelli, 81100 Caserta, Italy; vincenzo.nigro@unicampania.it (V.N.); annalaura.torella@unicampania.it (A.L.T.); 8Department of Neurosciences, Rehabilitation, Ophthalmology, Genetics, Maternal and Child Health, University of Genoa, 16147 Genoa, Italy; marcelloscala87@gmail.com (M.S.); strianop@gmail.com (P.S.); 9Paediatric Neurology and Muscular Disease Unit, IRCCS Istituto Giannina Gaslini, 16147 Genova, Italy; mariastellavari@gaslini.org; 10IRCCS Istituto delle Scienze Neurologiche di Bologna, UOC Neuropsichiatria dell’età Pediatrica, 40138 Bologna, Italy; anna.fetta@studio.unibo.it (A.F.); veronica.dipisa@aosp.bo.it (V.D.P.); ducciomaria.cordelli@unibo.it (D.M.C.); 11U.O. Genetica Medica, IRCCS Azienda Ospedaliero-Universitaria di Bologna, 40138 Bologna, Italy; francesca.montanari2@studio.unibo.it; 12Clinical Neurophysiology Unit, Scientific Institute IRCCS E. Medea, 23842 Bosisio Parini, Italy; robiepifanio4@gmail.com; 13Epilepsy and Clinical Neurophysiology Unit, IRCCS E. Medea Scientific Institute, 31015 Conegliano, Italy; paolo.bonanni@lanostrafamiglia.it; 14Molecular Biology Laboratory, IRCCS E. Medea Scientific Institute, 23842 Bosisio Parini, Italy; roberto.giorda@lanostrafamiglia.it; 15Child Neuropsychiatry Unit, Department of Medicine, Surgery and Dentistry, University of Salerno, 84084 Fisciano, Italy; opertofrancesca@gmail.com (F.O.); graziapastorino@gmail.com (G.P.); 16Department of Pediatric Neurology, Gazi University Faculty of Medicine, 06500 Ankara, Turkey; sarigeciliesra@gmail.com (E.S.); esra.serdaroglu@gmail.com (E.S.); 17Department of Pediatric Neurology, Mersin University, Mersin 33110, Turkey; cetinokuyaz@gmail.com; 18Department of Medical Genetics, Balcali Clinics and Hospital, Faculty of Medicine, Cukurova University, Adana 01330, Turkey; sevcantb@gmail.com; 19Medical Genetics Unit, Institute for Maternal and Child Health—IRCCS “Burlo Garofolo”, 34137 Trieste, Italy; luciana.musante@burlo.trieste.it (L.M.); flavio.faletra@burlo.trieste.it (F.F.); 20Child Neuropsychiatry Unit, Institute for Maternal and Child Health—IRCCS "Burlo Garofolo", 34137 Trieste, Italy; caterina.zanus@burlo.trieste.it; 21Rare and Complex Epilepsy Unit, Department of Neuroscience, Bambino Gesù Children’s Hospital, IRCCS, 00165 Rome, Italy; alessandro.ferretti@opbg.net; 22Neuroscience Department, Bambino Gesù Children’s Hospital, IRCCS, Full Member of European Reference Network EPICARE, 00165 Rome, Italy; federico.vigevano@opbg.net

**Keywords:** Pobinds, neurodevelopment, epilepsy, seizure, *CSNK2B*

## Abstract

Background: Poirier–Bienvenu Neurodevelopmental Syndrome (POBINDS) is a rare disease linked to mutations of the *CSNK2B* gene, which encodes for a subunit of caseinkinase *CK2* involved in neuronal growth and synaptic transmission. Its main features include early-onset epilepsy and intellectual disability. Despite the lack of cases described, it appears that POBINDS could manifest with a wide range of phenotypes, possibly related to the different mutations of *CSNK2B*. Methods: Our multicentric, retrospective study recruited nine patients with POBINDS, detected using next-generation sequencing panels and whole-exome sequencing. Clinical, laboratory, and neuroimaging data were reported for each patient in order to assess the severity of phenotype, and eventually, a correlation with the type of *CSNK2B* mutation. Results: We reported nine unrelated patients with heterozygous de novo mutations of the *CSNK2B* gene. All cases presented epilepsy, and eight patients were associated with a different degree of intellectual disability. Other features detected included endocrinological and vascular abnormalities and dysmorphisms. Genetic analysis revealed six new variants of *CSNK2B* that have not been reported previously. Conclusion: Although it was not possible to assess a genotype–phenotype correlation in our patients, our research further expands the phenotype spectrum of POBINDS patients, identifying new mutations occurring in the *CSNK2B* gene.

## 1. Introduction

Poirier–Bienvenu Neurodevelopmental Syndrome (POBINDS) is a rare disorder associated with mutations involving the gene *CSNK2B*. To our knowledge, only 50 cases have been reported in the literature. However, current evidence suggests that POBINDS could manifest with a wide range of symptoms, including early-onset epilepsy, a variable degree of intellectual disability (ID), developmental disability (DD), and growth abnormalities [1,2,3]. Such symptoms define a wide spectrum of phenotypes, ranging from well-controlled seizure and mild ID to intractable epilepsy, recurrent refractory status epilepticus (SE), and severe neurodevelopmental delay.

This evidence allows us to presume that the different clinical pictures might be related to the diverse alterations involving the POBINDS gene. *CSNK2B*, located in chromosome 6p21.33, together with *CSNK2A1* and *CSNK2A2*, forms casein kinase 2 (*CK2*), a ubiquitous serine/threonine kinase complex that takes part in various signaling pathways, including NF-kB, PTEN/PI3K/Akt and Wnt/b-catenin [4,5]. The literature has reported several alterations occurring in the β subunit of *CK2*, including missense, frameshift, nonsense, duplication, start loss, and splice site variants [1,3,6]. Different authors have hypothesized a possible link between the type of mutations and the clinical phenotype, with less severe forms of POBINDS that have been observed in patients with alterations in the zinc-finger region [1,3].

Interestingly, *CK2* appears to be abundant in the brain [7], where it takes part in the generation and development of neuronal processes, and plays a pivotal role in synaptic transmission [8]. *CK2* also appears to be involved in several neurodegenerative disorders, such as Alzheimer’s disease, Parkinson’s disease, and amyotrophic lateral sclerosis [9]. However, how alterations of *CSNK2B* lead to epilepsy is still not understood.

The purpose of this study is to achieve a better comprehension of the different POBINDS phenotypes and their relation to the mutations of *CSNK2B* through the analysis of a multicentric, Italian cohort of patients.

## 2. Materials and Methods

Our multicentric, retrospective study enrolled a total of 9 patients with childhood-onset epilepsy and ID/DD, recruited from ninecenters in Pisa, Bologna, Roma, Trieste, Catania, Lecco, Conegliano, Genova, and Salerno between October and December 2021.

Detailed clinical, laboratory, and imaging data were collected from corresponding clinicians through the filling of a specific form (see Appendix A). In order to confirm the clinical diagnosis of intellectual disability, and to assess its severity, different scales were used in consideration of patients’ ages. The Griffiths Mental Development Scale (GMDS) was employed, as well as the Leiter-R evaluation. To define the degree of severity, reference is made to IQ, according to ICD-10, which identifies four severity levels such as mild (IQ: 50–69), moderate (IQ: 20–34), and profound (IQ < 20).

Written, informed consent was obtained from all parents, and from patients if age of consent was reached.

CSKN2B variants were identified by epileptic encephalopathies next-generation sequencing panels or clinical whole-exome sequencing.

Variants were analyzed in order to define their pathogenicity and to assess whether they were nonsense, frameshift, missense, missense in frame, or splicing variants. This classification was performed using Varsome (https://varsome.com, accessed on 21 December 2021) [10]. Each mutation was therefore matched through ClinVar to determine if any of them have previously been published.

For the recruitment process, the following inclusion criteria were adopted: age below 18 years, CSKN2B mutation detected through genetic analysis, and availability of follow-up data. Exclusion criteria involved other eventual mutations likely to be pathogenic, incomplete data, and lack of informed consent.

Of the 10 initial selected patients, one was excluded from the study as a carrier of an SCN1A mutation, which was likely pathogenic in other coding sequences.

For each participant seizure, type at onset was described, together with eventual further seizure types and their correlation with the electroencephalographic recording.

Our research was focused on neurological and motor development impairments, as well as on body dysmorphisms and endocrinological, cardiac, and other system abnormalities, aiming to achieve a better definition of the phenotypes spectrum and to identify a significant correlation between genotype and phenotype.

Quantitative variables were tested for normality using the Shapiro–Wilk Test and are expressed as mean ± standard deviation (SD) or median and interquartile range (IQR), whereas qualitative variables are given as percentages. For association studies, we used the Wilcoxon test. A *p*-value of 0.05 was considered significant.

## 3. Results

A total of nine individuals with *CSNK2B* variants were investigated in this study. One patient has been reported elsewhere [11]. Concerning the previously published case, we collected additional data from the follow-up, therapy, and epileptic phenotype.

Table 1 summarizes the main characteristics and clinical features of our patients. Our cohort included three (33%) males and six (67%) females. All of them presented epilepsy and a different degree of intellectual disability. The median age at onset was 10 months (range: 0–65 months).

All patients presented de novo mutations of *CSNK2B*, which were missense in four (44%) cases, frameshift and nonsense in two (22%) cases, and splicing in one (11%) case (Figure 1), as shown in Figure 1.

Only three variants were previously described in the literature: the mutation presented by patient 2 (NM_001320.6:c.108dup) has been reported by Sakaguchi et al., whereas Yang et al. and Ernst et al. detected the variants discovered in patient 3 (NM_001320.6:c.494A > G) and patient 5 (NM_001320.6:c.368–2A > G), respectively. All the other mutations reported in our cohort have not been previously described.

Six out of nine cases showed generalized tonic–clonic seizures (67%), whilst one patient presented focal seizures, another experienced febrile convulsions, and one showed absences. We collected details about the treatment of seven patients out of nine. Valproate (VPA) was the most commonly used drug among seizure-free patients (patients 1, 2, 5, 6, and 9). Five patients became seizure free after monotherapy (patients 1, 5, 6 and 9 with VPA, patient 3 with Levetiracetam). Two patients became seizure free after polytherapy (patient 2 with LEV and VPA, patient 8 with Ethosuximid, and then Lamotrigin and Zonisamide).

Five (56%) cases displayed only mild intellectual disability, whereas patients 3, 6 and 7 showed a profound ID (Figure 2). Interestingly, only patient 5 did not present any speech difficulty, which was observed in 89% of cases, with patients 3 and 8 presenting mutism.

The other abnormalities observed included autistic features, occurring in two (22%) patients, short stature, which was reported in patient 3 and patient 8, and facial dysmorphism displayed in six (67%) patients (such as a triangular-shaped face, small nose, horizontal eye rims, hypertelorism, sparse eyebrows, depressed and receding forehead, prominent glabella, small and round ears, long and prominent filter or, on the contrary, a short filter with small teeth that have undergone diastasis, arched upper lip with downwards angles, retrognazia or prognathism, and a depressed nasal bridge) (Figure 2). Interestingly, three patients (33%) presented lymphatic and blood vessel abnormalities, namely lymphedema in patient 6, vascular skin anomalies in patient 8, and a cavernous angioma on the ear lobe in patient 9.

All patients underwent electroencephalogram (EEG) at onset, which displayed different discharges in all cases.

Focal, multifocal, or generalized discharge was recorded with a spike and spike and wave complex. Only one patient showed some diffuse slow waves discharge, and another patient presented a burst suppression pattern (Figure 3).

Magnetic Resonance Imaging (MRI), on the other hand, was performed in the cases of eight patients, and returned normal results in five of them (62.5%). Patient 3 presented hypoplasia of the cerebellar worm, delayed myelination, and increased size of cisterna magna. Patient 6 displayed adipose transformation of the Filum Terminale, a colloid cyst of the third ventricle, hypoplasia of the corpus callosum and pons, and enlargement of the cerebrospinal fluid spaces (Figure 4). Finally, patient 7 showed gyral simplification and delayed myelination.

### 3.1. PATIENT 1

Patient 1 is a 16-month-old girl with an unremarkable history experienced a cluster of several non-prolonged focal seizures and subsequent generalized tonic–clonic seizures (GTCS), each one lasting for approximately 2 to 5 min. The cluster needed intravenous midazolam and phenobarbital for its interruption. She was given Sodium Valproate (VPA) of up to 20 mg/kg, and exhibited good seizure control until 2 years of age, when she experienced a cluster of three seizures that required an increase in dosage to 25 mg/kg. Ictal EEG highlighted a left temporo-posterior focus with subsequently generalized spike–wave discharges. The interictal EEG showed a left temporo-posterior focal discharge (Figure 2). The brain MRI was found to be negative. Currently, at the age of 2.5 years, she shows a mild ID with speech delay, and is able to use single words. She has no motor delay but presents a mild hyperactivity disorder. No facial dysmorphism was observed, The epileptic encephalopathies NGS panel showed that she is affected by a heterozygous variant on exon 3, which determines a stop codon (NM_001320.7:c286c > T, p.Gln96*).

### 3.2. PATIENT 2

The second patient is a boy with epilepsy and mild ID. The seizure type was generalized tonic-clonic seizure with a frequency of 2–3 times daily at the time of onset (20 months of age). A further seizure type observed was focal seizure. He was treated with VPA with good control. EEG showed multifocal spikes at onset and diffuse sharp waves at follow-up. He communicates through short sentences, shows autistic features, and presents facial dysmorphisms, such as a triangular-shaped face, hypertelorism, short filter, and sunken orbital region. He presented a heterozygous frameshift variant on exon 1 that results in a premature stop codon (nm_001320.7:c.108dup, p.Thr37TyrfsTer5).

### 3.3. PATIENT 3

This patient showed signs of epilepsy at 7 days old and a profound intellectual disability. The seizure type was focal-clonic seizure and epilepsy was under control with Levetiracetam (LEV) treatment. The EEG at onset was abnormal with focal anomalies, mostly on the right side. EEG at follow-up showed paroxysmal multifocal activity. The brain MRI showed hypoplasia of the cerebellar worm, delayed myelinization, and a megacisterna magna. Severe psychomotor impairment was recognized. He presented poor motor skills, hypotonia, and distal dystonia. He displays facial dysmorphism, such as a triangular-shaped face, small nose, horizontal eye rims, and small and round ears. Other features observed included laryngomalacia, hypoacusia, and growth restriction.

The patient was found to have an already-known heterozygous missense variation on exon 5 (nm_001320:c.494A > G,p.His165Arg).

### 3.4. PATIENT 4

Patient 4 is a 7-month-old boy with epilepsy and mild ID. The type of seizure at onset was generalized tonic-clonic seizure and it presented as clustered seizures. Further types were focal seizures. Epilepsy was well controlled with VPA treatment, which commenced at onset with up to 20 mg/kg. The EEG showed multifocal spikes and generalized sharp waves. The brain MRI was normal. He displayed normal motor developmental milestones. Currently, at the age of 15, he has a mild speech and behavioral impairments. No facial dysmorphism was observed.

He is affected by a de novo heterozygous mutation on exon 1, which is of the nonsense type (nm_001320:c.27G > A, p.Trp9Ter).

### 3.5. PATIENT 5

Patient 5 is a 10-month-old girl who developed epilepsy and mild ID. The seizure type at onset was GTCS and presented with a frequency of three episodes in 2 months, before starting treatment with valproate. She relapsed during an attempt to reduce treatment at around 7 years of age. The EEG at onset was normal. The follow-up EEG showed some diffuse slow waves during sleep and drowsiness. The brain MRI was found to be normal. She has mild attention and memory impairments and does not present any dysmorphic traits.

She was found to be affected, through clinical whole-exome sequencing, by a heterozygous splicing variant (nm_001320:c.368–2A > G) on exon 4 that has been mentioned previously in the literature.

### 3.6. PATIENT 6

Patient 6 is an 18-year-old girl with profound intellectual disability, motor delay, and severe hypotonia. She started to walk at age 4. She said her first words at 5 years of age and communicates only through short sequences. At 8 months old, she presented febrile seizures and was treated with valproate for 9 years with good seizure control. The brain MRI showed low-lying spinal cord with adipose transformation of the Filum Terminale, colloid cyst of the third ventricle, microcephaly, hypoplasia of the corpus callosum and pons, and enlargement of the cerebrospinal fluid spaces. Dysmorphic features included a depressed and receding forehead, hypertelorism, prominent glabella, low-placed ears, short filter, small and teeth that have undergone diastasis and retrognathia (Figure 1). Other findings included lymphedema, severe scoliosis, long tapered fingers, and cardiac anomalies such as IM and IT.

She showed a heterozygous missense in the frame variant on exon 2, which causes the deletion of only one amino acid (nm_001320:c.181_183del, p.Glu61del, p.Glu61del).

### 3.7. PATIENT 7

Patient 7 was a 16-month-old girl with epilepsy and profound ID/DD. Seizure type at onset was GTCS. She also developed infantile spasms. The EEG at onset showed focal and multifocal discharges with burst suppression. She has severe spastic hypertonus and motor delay, as well as speech delay. The brain MRI highlighted gyral simplification and delayed myelination. She presents dysmorphic features such as a high palatal arch and microcephaly.

She is affected by a heterozygous variation on exon 4, which was found to be missense (nm_001320: c.332G > A, p.Arg111His).

### 3.8. PATIENT 8

Patients 8 is a girl who developed childhood absence epilepsy (CAE) at the age of 5. Her seizures appeared daily at onset, and the EEG showed typical spike–wave complexes (3 Hz). Further episodes were focal with autonomic signs and awareness impairment. Her starting treatment was ethosuximid, but was interrupted after 16 months and replaced by lamotrigin (2 mg/kg/die) and zonisamid (20 mg/kg/die) due to gastrointestinal side effects and treatment discontinuity. The EEG at follow-up highlighted multifocal epileptiform discharges (Figure 5). The patient performed an MRI, which displayed Chiari type 1 malformation and syringomyelia. She has a mild ID. Her motor skills are normal, but she is not able to communicate, presenting language delay and selective mutism. She presents facial dysmorphisms such as broad forehead, frontal bossing, sparse eyebrows, hypertelorism, depressed nasal bridge, pointed chin, and prognathism. A growth delay has been discovered, along with vascular skin abnormality and a butterfly vertebra.

The clinical picture appears to be caused by a missense mutation on exon 1 (nm_001320:c.116T > G, p.Leu39Arg).

### 3.9. PATIENT 9

Patient 9 is a girl who developed epilepsy at 9 months old. Seizures initially appeared as a loss of contact, staring, and hypotonus, lasting less than 2 min. The episodes initially occurred weekly, and later she also developed GTCS. Epilepsy was well controlled with VPA treatment, which was started at onset with up to 20 mg/kg and continued until the age of 5. EEG at onset showed diffuse anomalies and rare slow waves (sn > dx), but turned normal after therapy introduction. The brain MRI was normal.

She started walking at 17 months old and said her first words around the same age. She suffers from different facial dysmorphisms, such as brachycephaly, a sunken orbital region, cavernous angioma on an earlobe, and head circumference at 10° centile. Finally, she is affected by a de novo heterozygous frameshift variant on exon 4, which causes a premature stop codon (nm_001320:c.384_394del, p.Met132LeuFs*110).

### 3.10. Isoform References

Isoform references: ENST00000375882 (NM_001320) (Figure 3).

## 4. Discussion

POBINDS is a rare neurodevelopmental disorder related to loss-of-function mutations of *CSNK2B*. It was first described by Poirier at al. in 2017 in two patients with intellectual disability and, in one case, myoclonic epilepsy [2]. Its pathogenesis appears to be strictly linked with dysfunctions of kinase *CK2*, a heterotetramer constituted by *CSNK2B*, *CSNK2A1* and *CSNK2A2* which is involved in several signaling pathways [5,9].

CK2 appears to play a pivotal role during embryogenesis and cell differentiation [12]. Nonetheless, it is also involved in CNS development, through its activity in transcriptional pathways [13].

Due to the merging features of this syndrome that have been recently described by several authors, our acknowledgments on the clinical impact of *CSNK2* mutations are constantly increasing. However, the wide spectrum of phenotypes described has made the diagnosis and clinical follow-up progressively challenging for clinicians.

The majority of POBINDS patients presented in the literature seem to exhibit epilepsy as a main feature, associated with a different degree of intellectual disability. However, the severity of phenotype is highly variable and to date a clear genotype/phenotype correlation has not been established yet.

Our cohort reported nine patients with different *CSNK2B* mutations. Age at onset did not differ from the data presented in the literature (median: 10 vs. 5.5 months; *p*-value: 0.09) [1,3,6,14,15,16]. Interestingly, all our patients displayed epilepsy, with generalized tonic–clonic seizure being the most common phenotype (67%), in agreement with previous studies [1,3,6,14,15,16]. Such findings could possibly be related to the diffuse abundance of *CK2* subunits within the brain, which might also act indirectly through the dysregulation of gated potassium channels [8].

All our patients except one (patient 9) show mild to severe intellectual disability; however, patient 9 is still too young to establish the presence of developmental delay. Of note, Li et al. reported three cases with normal cognitive status [1]. Speech impairments are also a common finding in the case of *CSNK2B* loss of function, occurring in 66% to 100% of patients [1,3,6,14]. In our cohort, only patient 5 had no speech difficulties; absence of speech, on the other hand, occurred in two cases (22%).

Several authors reported different types of mutation of *CSNK2B*. To the best of our knowledge, the most common were missense variants (13 cases) [3,14], followed by nonsense (seven cases) [3,15], splicing and frameshift, reported in five cases each [3,6,14], and start loss variants (two cases) [3]. Similar results were detected in our cohort, where missense mutations occurred in 33% of cases. Interestingly, six of our patients (1, 4, 6, 7, 8 and 9) presented variants that have never been reported elsewhere, representing a further expansion of our knowledge on the genetic mechanisms that lead to *CSNK2B* loss of functions.

Merging evidence has recently shown that dysmorphism represents a common finding in POBINDS patients, reported in 12 cases by Ernst et al. [3], and in one patient by Nakashima et al. [16]. Our results confirmed such findings, with five (55%) patients presenting facial dysmorphism, as shown in Figure 2. Abnormalities of cranial size have also been described by Ernst et al., who reported two patients with microcephaly and four patients with macrocephaly [3]. In our cohort, 33% of cases presented microcephaly.

The ubiquitous activity of *CK2*, involved in myogenic and osteogenic differentiation [12] jointly with its abovementioned abundance in the brain [7], could be the main explanation for the endocrinological abnormalities observed in POBINDS patients. Yang et al. reported two cases with height below two SD from average [1], while Ernst et al. described a boy with growth hormone deficiency and a girl with precocious puberty [3], which was subsequently assessed by Nakashima et al. [16]. Consistently with these findings, our analysis detected two patients with growth delay.

Interestingly, the MRI showed three abnormal cases. Two of them showed delayed myelination, which was also observed in one patient by Li et al. [1]. This finding could be explained by the role of *CK2* in brain development [12,13], and perhaps even in myelinogenesis. One patient presented syringomyelia and Chiari 1 syndrome, which has not been previously reported in association with POBINDS

Finally, three patients of our cohort showed lymphatic or blood vessel abnormalities, which could represent another feature of the impaired function of *CK2* during embryogenesis.

In the majority of reported cases, the overall response to anti-seizure medications appears to be effective, generally with the employment of two drugs. To our knowledge, the most common anti-seizure medications (ASM) employed in POBINDS patients have been levetiracetam and sodium valproate [1,3,6,14,15,16]. In our cohort, VPA was the main drug used, often in monotherapy (Figure 3) Only one patient relapsed on discontinuation of treatment. The other ASMs employed were levetiracetam, ethosuximide, lamotrigin and zonisamide, which are often in association with each other [1,14]. Despite the lack of information on all patients, we were able to assess the effectiveness of treatment.

In conclusion, we reported nine unrelated patients with a de novo heterozygous mutation in the *CSNK2B* gene; six of them showed new variants that had not been reported elsewhere. Our research further expands the phenotype spectrum of mutations for POBINDS. Unfortunately, in analyzing our cohort of patients it was not possible to identify a correlation between genotype and phenotype. In particular, in analyzing ID or epilepsy, the variants expressed are different even when they have the same phenotype. For example, patients with mild ID reported mutations of all types (missense, frameshift, and nonsense). In our cohort, the severity of the intellectual disability may not only be related to the type of variant, but to a correlation of the exon involved. Four of our patients with milder phenotypes presented mutation in exons 2 and 3, respectively; however, these data need further investigation with a bigger cohort of patients.

*CSNK2B* protein is ubiquitously expressed and involved in various cellular processes, but its role in CNS, other organs, and system development needs to be further researched, including animal models.

Further studies are therefore mandatory to better delineate the disease spectrum and its mechanisms of action, and to enhance our understanding of the genotype–phenotype correlation [17]. Nonetheless, we believe that due to the lack of genetic exams analyzing *CSNK2B* and the different phenotypes showed by patients, the real prevalence of POBINDS could be underestimated. The employment of WES with a broader range of genes analyzed could possibly allow us to perform a more precocious diagnosis and a better follow-up of patients.

## Data Availability

Not applicable.

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
