# Peer review of "Expanding Phenotype of Poirier–Bienvenu Syndrome: New Evidence from an Italian Multicentrical Cohort of Patients"

_genes, 2022, doi:10.3390/genes13020276_

Round 1

Reviewer 1 Report

The authors describe 9 patients with Poirier-Bienvenu syndrome, due to different mutations in the CSNK2B gene.

Reading the manuscript is a little difficult because a lot of words are stuck together in the text.

Table 1 is also difficult to read. I would suggest another format to give a better overview.

In Graphic 1 and 2 I do not understand why there are two extra collums for female and male patients. It gives no extra information, and the collum "Literature" represents female and male patients. The condition is due to a gene on chromosome 3, no reason for difference between female and male patients I think.

In Graphic 3 there is a remarkable difference in therapy between female and male patients, but the authors give no comment on this finding. Is there a reason for this difference? 

results: line 127: Give numbers: 6 out of 9 patients showed generalized.... and NOT "The overwhelming majority....", there are only 9 patients described.

line 263: ...Seizures?.... ???? 

Author Response

Reviewer #1 comments:

We thank the Reviewer for her/his careful reading of the manuscript and the insightful suggestions

  • Reading the manuscript is a little difficult because a lot of words are stuck together in the text.

Reply: We apologize for such inconvenient, which was due to the different software used by our Authors. We corrected each merged word found within the text.

  • Table 1 is also difficult to read. I would suggest another format to give a better overview.

Reply: We thank for the suggestion; table’s format has been changed in order to ease the reading process.

  • In Graphic 1 and 2 I do not understand why there are two extra collums for female and male patients. It gives no extra information, and the collum "Literature" represents female and male patients. The condition is due to a gene on chromosome 3, no reason for difference between female and male patients I think.

Reply: The subdivision between “male” and “female” aimed to display an eventual difference between the two genders. However, such difference was not detected. Nonetheless, Table 1 contains as well the necessary information regarding the type of mutation and the degree of intellectual disability of our patients. Therefore, we agreed with the Reviewer and removed the columns from our Graphics.

  • In Graphic 3 there is a remarkable difference in therapy between female and male patients, but the authors give no comment on this finding. Is there a reason for this difference? 

Reply: The gender difference of treatment initially detected in our cohort was due to the lack of information on the therapy adopted by three patients. We have collected the missing data and further updated our graphic, which showed no statistical difference between the two populations. However, we found interesting that in our cohort a greater success rate was achieved through the administration of Valproic Acid, rather than Levetiracetam, compared to available literature.

  • results: line 127: Give numbers: 6 out of 9 patients showed generalized.... and NOT "The overwhelming majority....", there are only 9 patients described.

Reply: We have modified the sentence by removing “The overwhelming majority” and changed it with the ratio of patients showing tonic-clonic seizures (line 127).

  • line 263: ...Seizures?.... ????

Reply: Such quote was accidentally left there from a precedent draft of the manuscript;  we apologize for the inconvenient and removed it.

Reviewer 2 Report

The author did retrospective study recruited 9 patients with POBINDS, detected 39 through Next Generation Sequencing panels and Whole Exome Sequencing. This is article could be adding a new perspective about that disease. The sequencing result better presented in figure.

Author Response

We thank the Reviewer for his/her comments and hope that, by increasing our knowledge on POBINDS, our work will encourage further researches on this disease.

Reviewer 3 Report

The ms. about extremely rare Poirier-Bienvenu Neurodevelopmental Syndrome (POBINDS) provides important information about the neurophysiology related to CSNK2B gene alternations.

However, few corrections should be made.

The information about patient’s phenotype should described in more details and following the same format for all patients. For example, for patient 3 the age of the patient is not stated. It might hardly be 7 days, as the diagnosis of ID is mentioned. I would like the authors to elaborate more about the way clinical phenotype is collected. What kind of form clinicians are filled? What instruments were used to diagnose ID, etc?

Table 1 hard to read as all info is very tight – probably enlargement of the column by orienting the table horizontally will help.

Better representation of EEG abnormalities is needed. First of all, the scales (time and amplitude) should be added to the figure 1. Secondly, if more EEG data is available with clear atypical patterns, they might also be added for representation.

In the results it says that 50% of patients of 8 sowed abnormal MRI, but only 3 patients is described.

Several abbreviations should be explained, e.g. «recurrent refractory SE»

There are many places where several words are merged together. This should be examined and corrected throughout the ms.

Author Response

Reviewer #3 comments:

We thank the Reviewer for his/her work and further updated our manuscript in order to meet his/her expectations.

  • The information about patient’s phenotype should described in more details and following the same format for all patients. For example, for patient 3 the age of the patient is not stated. It might hardly be 7 days, as the diagnosis of ID is mentioned. I would like the authors to elaborate more about the way clinical phenotype is collected. What kind of form clinicians are filled? What instruments were used to diagnose ID, etc?

Reply: We have further specified the clinical, laboratory and imaging data of our patients. Furthermore, we described all cases through the same format, in order to ease the reading and the comparison of patients.

As explained in the manuscript, we have collected our data through a written form that has been filled by each centre (line 86), which you will find attached to this reply. Such form could be added as well as Supplementary Material.

Regarding the assessment of Intellectual Disability, we provided a better explanation on the methods employed by adding the following: “In order to confirm the clinical diagnosis of Intellectual Disability and to assess its severity, different scales were used in consideration of patients’ age. The Griffiths Mental Development Scale (GMDS) was employed, as well as the Leiter-R evaluation. To define the degree of severity, reference is made to IQ, according to ICD-10, that identifies four severity levels such as mild (IQ: 50-69), moderate (IQ: 20-34), and profound (IQ < 20).” (lines 86-90).

  • Table 1 hard to read as all info is very tight – probably enlargement of the column by orienting the table horizontally will help.

Reply: We thank for the suggestion, table’s format has been changed in order to ease the reading process.

  • Better representation of EEG abnormalities is needed. First of all, the scales (time and amplitude) should be added to the figure 1. Secondly, if more EEG data is available with clear atypical patterns, they might also be added for representation.

Reply: We have added eeg data about patient (8) in figure 4. We added in the figure 1 legend time and amplitude of the eeg recorded (100 Hz, 30 sec) and modified the text.  

  • In the results it says that 50% of patients of 8 sowed abnormal MRI, but only 3 patients is described.

Reply: We have obtained further details regarding MRI of missing patients, which have been added to our manuscript. MRI resulted abnormal in 3 out of 8 patients. Therefore, we modified line 152 with the following: “[…] and resulted normal in 5 of them (62.5%)”.

  • Several abbreviations should be explained, e.g. «recurrent refractory SE»

Reply: We have specified the meaning of the abbreviations that were not previously explained, such as “SE” (Status Epilepticus), “DD” (Developmental disability), “MRI” (Magnetic Resonance Imaging), “ASM” (Anti-Seizure Medication) and “EEG” (Electroencephalogram)

  • There are many places where several words are merged together. This should be examined and corrected throughout the ms.

Reply: We apologize for such inconvenient, which was due to the different software used by our Authors. We corrected each merged word found within the text.
